# Understanding Mothers’ Worries about the Effects of Disaster Evacuation on Their Children: A Cross-Sectional Study

**DOI:** 10.3390/ijerph20031850

**Published:** 2023-01-19

**Authors:** Chisato Yamazaki, Hisao Nakai

**Affiliations:** School of Nursing, Kanazawa Medical University, 1-1 Uchinada, Kahoku 920-0265, Ishikawa, Japan

**Keywords:** disaster preparedness, childcare center, allergy, anxiety about evacuating, grab bag for disaster risk reduction

## Abstract

In Japan, there is an imminent threat of major earthquakes and floods. Children’s health is increasingly at risk from climate-change-related disasters. The purpose of this study was to identify factors related to mothers’ worries about the effects of evacuation on their children. Participants were mothers whose children attended a childcare center in one municipality in Ishikawa, Japan. A cross-sectional design was used. A questionnaire was developed based on previous studies, and it was used to conduct a survey. A total of 1298 individuals who provided valid data were included in the analysis. The following factors were related to mothers’ worries about the effects of evacuation on their children: not having prepared a grab bag as a disaster risk reduction strategy, having no neighbors to help them in case of disaster, having children aged <3 years, and having children with allergies. The mothers of children <3 years old with allergies who are unprepared and have no social support are likely to worry about evacuating their children. Policymakers must be aware that the mothers of children aged <3 years and the mothers of children with allergies experience substantial concerns about the effects of evacuation on their children.

## 1. Introduction

In recent years, the imminent threat of a Nankai Trough earthquake or an earthquake directly under the Tokyo metropolitan area has become apparent in Japan. The Noto Peninsula Earthquake in March 2007 had a seismic intensity of 6 and killed one person [1]. In December 2021, an earthquake swarm caused by fault weakening due to pore fluid migration into the fault zone was detected in the Noto Peninsula [2]. In June 2022, an earthquake that measured a low 6 on the Japanese Seismic Intensity Scale occurred in Suzu City, Ishikawa Prefecture [3], and it was believed to have been caused by fault weakening. In addition, it has been predicted that localized heavy rainfall and an increase in the number of typhoons triggered by global warming will cause substantial future damage [4]. In Japan, heavy rainfall caused by Senjo-kousuitai, a band-shaped precipitation system, frequently occurs from June to September, causing serious damage [5,6]. Senjo-kousuitai is less likely to occur on the side of the country bordering the Sea of Japan [7], but in August 2022, heavy rainfall accompanied by Senjo-kousuitai occurred in Ishikawa and caused major damage. Approximately 1800 people were evacuated due to large-scale flooding and landslides caused by overflowing rivers [8].

Children’s health is increasingly at risk from recent climate-change-related disasters [9]. During the 2011 Great East Japan Earthquake and tsunami, the parents of children with food allergies experienced difficulties in obtaining allergen-free ingredients and powdered milk, and there were many reports of atopic dermatitis exacerbated by the inability to bathe [10,11]. Although government recommendations state that mothers should stockpile allergen-free ingredients and allergy medication, a 2018 survey showed that only approximately 30% of mothers did so [12]. Mothers and children are particularly vulnerable during disasters [13,14]. This is because disaster exposure increases children’s risk of acute illness, developmental disabilities, and being underweight [15]. In addition, maternal anxiety affects children. A post-Hurricane Iruma study showed that evacuation itself may contribute to the onset of anxiety and depression in mothers and children [13], and a post-Hurricane Katrina study highlighted the need for mental health care services for children many years after the event [16]. Therefore, post-disaster social and psychological support for children and parents is very important [17,18]. It has long been noted that there is an interaction between the anxiety of mothers and that of their children [19] and that mothers have a particularly large effect on their children [20]. Because the mother’s anxiety is transmitted to and affects their child, it is essential that mothers learn to cope with anxiety [21].

Parental anxiety, mental instability, a poor family environment, irritability, and violence in post-disaster evacuation settings increase stress transmission to children [22] and increase the risk of detrimental effects on the child’s mental health. In a survey conducted after the 2011 Van earthquakes in Turkey, mothers who were affected by the earthquakes and forced to migrate showed not only increased maternal aggression towards their children due to psychological distress caused by the environmental changes of migration but also a lack of positive emotional and physical engagement. This may have led to destructive behavior in the children [23]. That is, mothers’ feelings of anxiety after a disaster are transmitted to their children and have a negative effect on them. It is therefore important that mothers are able to cope with and manage their feelings of anxiety. Some research shows that parental adjustment, a depressed mood in the home, irritability, violence, and less support for children are predictive factors for children’s recovery from anxiety symptoms and fears after a disaster [24].

There have been several studies of parental anxiety about evacuating with children during a disaster. For example, after the Great Hanshin-Awaji Earthquake, the parents of medically vulnerable children expressed concerns about the availability of medical treatment at evacuation centers. During the Great East Japan Earthquake, the parents of children with disabilities avoided evacuating to public shelters because they worried that their children would become confused and disturb others with disruptive behaviors, such as screaming [25]. The parents of children who used medical equipment, such as artificial respirators, expressed their intention not to evacuate to public shelters due to the lack of space and concerns about the availability of medical equipment and medical care in the shelter [26]. One study found that evacuation shelter workers were concerned about their ability to effectively respond during an evacuation because they felt that the shelter environment was not appropriately equipped for children with allergies [27]. However, there have been no studies focusing on maternal anxiety about the effects of evacuation to shelters on nursery school children.

To ensure that mothers and children respond calmly and cooperatively to disaster evacuation, it is important to try to clarify and resolve any fears or concerns that mothers have during normal times. Preparedness strategies are central to coping with disasters, such as preparing disaster supply kits for children and families, participating in evacuation drills, and mutual support within neighborhoods [9,28,29,30]. In a study conducted after the 2015 South India floods, a lack of disaster preparedness during normal times caused substantial damage, leading to a recognition by parents, children, and other community members of the importance of disaster preparedness [31]. Parent and child disaster preparedness emphasizes the importance of children’s active participation in disaster preparedness [32]. Mutual learning involving children and parents can increase children’s disaster knowledge and risk perception [33]. In any disaster preparedness efforts for mothers and children, it is important to consider the mutual effects of the disaster on both parties, to address mothers’ feelings of anxiety, to increase parents’ knowledge and risk perception of disasters, and to ensure that parents prepare for disasters in normal times together with their children. In addition to encouraging vulnerable people to prepare for disasters themselves, there is a need to establish shelters specifically for this population [34,35]. However, the factors that affect mothers’ worries about disaster evacuation in relation to chronic mental and physical illness in their children are unknown. The objective of this study was to survey mothers with children who attend daycare facilities in order to (1) identify the prevalence and type of allergies, allergens, and disaster preparedness behaviors and (2) identify factors associated with maternal anxiety about the effects of disaster evacuation on children. Because it is likely that disasters caused by climate change will increase worldwide in the future, this is an important issue in the consideration of evacuation measures for children and mothers during disasters.

## 2. Design and Participants

### 2.1. Study Design and Participants

This was a cross-sectional study that used self-administered questionnaires. All parents whose children attended a childcare center in one municipality in Ishikawa Prefecture were included in the study. Childcare centers in Japan are defined according to the Child Welfare Act, and they are facilities that care for children aged 0 to 5 years at the request of their parents [36,37]. They are operated by a variety of entities, including local governments, social welfare corporations, joint-stock companies, and non-profit organizations [38]. The departments in charge of childcare facilities in the target municipality were asked to take part in the survey. An outline of the survey was then presented at a meeting of the heads of childcare facilities, and their cooperation was requested. The representatives of the childcare facilities verbally asked parents to cooperate in the survey, and they distributed the survey request documents, self-administered questionnaires, and envelopes to the parents. Parents were asked to drop their completed questionnaires into a collection box placed at the childcare facility. Consent for study participation was assumed if parents deposited their completed questionnaires into the box. The study period was from July 2019 to September 2019.

### 2.2. Data Collection

To assess mothers’ backgrounds, the respondents were asked their age, their length of residence, whether they were pregnant, and whether they had evacuation experience. The possible responses were “no” or “yes.”

To assess disaster preparedness and support, respondents were asked to reply “no” or “yes” to the following questions: Can you visualize yourself living in an evacuation center? Have you prepared a grab bag for disaster risk reduction? Do you have parents or relatives who can help you in case of disaster? Do you have neighbors who can help you in case of disaster? Are you worried about the effects of disaster evacuation on your children? The aim of this latter question was to assess concerns about evacuation to a public shelter in Japan in the event of an earthquake and/or flooding. As such, the intention was to measure a level of worry other than the normal low-level constant anxiety that mothers have about their children.

To assess children’s background and allergies, the respondents were asked about the number of children and the children’s sex, age, whether they were breastfeeding, eating baby food, allergic to any foods, or had any non-allergic conditions.

There were multiple response options for the question about food allergens. Parents were asked to select from chicken egg, milk, wheat, peanuts, buckwheat, shrimp, and crab and to note down any other allergies that their children had. The respondents were asked whether they used allergy-free foods (“no” or “yes”); whether they stockpiled allergy-free foods (“1–3 days, 1 week, more than 2 weeks, or no”); if their children took medication for allergies (“no” or “yes”); and if they stockpiled medication (“1–3 days, 1 week, more than 2 weeks, or no”). The respondents were asked to state if their children had atopic dermatitis (“no” or “yes”). If they responded “yes”, they were asked to select from moisturizing lotion or steroid ointment and to note down any other treatments that they used.

### 2.3. Analytical Methods

Of the participants, those who responded to all the items regarding the mother’s background, disaster preparedness and supporters, children’s background, and factors related to worries about children in case of a disaster were included in the analysis.

After the Great East Japan Earthquake, it was recognized that mothers who have to evacuate with their children are particularly vulnerable during disasters, and they require greater consideration during the evacuation process [39]. Although there is no clear information about the effects of maternal age and maternal perceptions of evacuation on coping with disasters, it is likely that childbirth, childcare, and other life experiences, rather than age per se, affect mothers’ experiences of evacuating with children. Therefore, considering the age distribution of the mothers in this study and that of the mothers of nursery school children in previous studies [40,41], we divided maternal age into two categories: 35 years or older and under 35 years.

The current years of residence were categorized as “less than 5 years” and “more than or equal to 5 years”; the number of children was categorized as “one” and “two or more” [42]; and children’s age was categorized as “less than 3 years” and “more than or equal to 3 years” with reference to the childcare guidelines of the Japanese Ministry of Health, Labour and Welfare. These guidelines provide recommendations for the provision of care at childcare facilities. The guidelines identify the period from birth to 3 years of age as an important period for the early development of a child’s mental and physical abilities; after 3 years, children experience further development in physical growth, interrelationships, and collaborative activities [43]. The distribution of the responses to each item in the group of respondents who answered “yes” to the question about children’s allergies was obtained. The distributions of the responses to the questions on stockpiling allergen-free food and allergy medicines in preparation for disasters were also obtained.

The chi-square test or Fisher’s exact test was performed to examine the relationship between mothers’ worries about the effects of disaster evacuation on their children, mothers’ background, disaster preparedness and supporters, and children’s background. A binomial logistic regression analysis was performed to examine factors associated with mothers’ worries about evacuating their children. The independent variables were tested using chi-square, and the following variables, which had a statistical significance of less than 5%, were selected as predictors: “not having prepared a grab bag for disaster risk reduction”, “having no neighbors to help in case of a disaster”, “having one child”, “having children aged under 3 years”, “breastfeeding”, “having children still eating baby food”, “having children with an allergy”, and “having children with an illness other than an allergy”. Each variable was checked for multicollinearity (coefficient of variance expansion ≥10) prior to performing a binomial logistic regression analysis. Maternal age was selected as a potential confounding variable. Variable input was performed using the forced input method. The Hosmer–Lemeshow test was used to assess the model fit. The significance level was set at 5%. SPSS Ver27 (IBM Corporation, Armonk, NY, USA) was used for all statistical analyses.

### 2.4. Ethical Considerations

This research was conducted in accordance with the Declaration of Helsinki, 1995 (as revised in Seoul, 2008), and it was carried out with the consent of the university medical research ethics review committees at the authors’ universities (No. I400). The participants’ freedom to refuse to participate, freedom to withdraw their cooperation, anonymity, and privacy protection were explained in writing, and consent was assumed by participants depositing the survey form in the collection box.

## 3. Results

Responses were obtained from 1908 parents (a 72.4% response rate), of which 1790 (93.8%) were mothers. Of these, 1298 (72.5%) provided valid responses for all survey items.

Of the participants, 798 (61.5%) reported anxiety about the effects of disaster evacuation on their children, and 583 (38.5%) did not. Data on the reported demographic characteristics of the participants are shown in Table 1.

Data on reported allergies and the stockpiling of food and medication are shown in Table 2.

Table 3 shows the results of the univariate analysis of the associations between mothers’ worries about the effects of evacuation on their children and other factors. There were significant differences in several variables between mothers who reported concerns about the effects of a disaster on their children and those who did not. For example, a higher percentage of mothers who expressed concerns reported not having prepared a grab bag for disaster risk reduction (*n* = 917, 88.8%; *p* = 0.01), having no neighbors to help in case of a disaster (*n* = 410, 67.3%; *p* = 0.00), having one child (*n* = 265, 68.8%; *p* = 0.00), having children under 3 years old (*n* = 400, 66.6%; *p* = 0.00), breastfeeding (*n* = 51, 77.3%; *p* = 0.00), having children who eat baby food (*n* = 56, 80.0%; *p* = 0.00), having children with some kind of allergy (*n* = 122, 78.2%; *p* = 0.00), and having children with an illness other than allergies (*n* = 38, 77.6%; *p* = 0.02).

Table 4 shows the results of the binomial logistic regression analysis, which identified several factors related to the dependent variable of mothers’ worries about the effects of evacuation on their children.

## 4. Discussion

The purpose of this study was to identify factors related to mothers’ worries about the effects of evacuation on their children. Almost two-thirds of participants expressed concerns about the effects of disaster evacuation on their children. The main findings of this study were that mothers were likely to show evacuation anxiety if they were unprepared for evacuation, had no social support, had children younger than 3 years, or had children with allergies. Policymakers need to be aware that the mothers of children younger than 3 years and the mothers of children with allergies experience anxiety about how evacuation may affect their children.

In a previous study of the mothers of children attending childcare facilities in Japan, the average age of mothers was 35.7 years [44], and in a parent questionnaire study in 2022, 50.1% were in their 30 s, 41.6% were in their 40 s, and 8.2% were in their 20 s [45]. Therefore, it is likely that the participants in the present study were representative of the population of the parents of children attending childcare centers in Japan. Regarding children with allergies, a survey of children at childcare facilities in Tokyo found that 6.3% of the children [46] had food allergies, which is approximately the same percentage as that in the present study. Therefore, the present data seem generally representative of this population. Regarding the prevalence of atopic dermatitis, according to a 2014 survey in Japan, the prevalence at 3 years old was 7.3% [47], and it was 12.2% in 2019 [48]. Of children diagnosed with atopic dermatitis by 2 years old, approximately 43% were cured by 3 years old, and approximately 38% had mild and recurrent exacerbations [49]. Considering the differences in incidence between the different age groups and the recurrent, distressing, and exacerbating nature of the disease, it is not possible to determine whether the present study’s population is representative of Japanese children with atopic dermatitis.

The present findings suggest that the mothers of children younger than 3 years old with allergies who have not prepared a grab bag for disaster risk reduction and have no one to rely on may be living with anxiety about the effect of evacuation on their children. It has long been reported that medically fragile populations are less prepared for disasters despite their concerns [50,51,52]. Similarly, some of the mothers of children with allergies in this study had not prepared a grab bag for disaster risk reduction and so may have experienced anxieties about the effect of a disaster on their children. Preparing an emergency grab bag is recommended worldwide as a disaster risk reduction strategy [53]. However, the number of recommended items and their beneficial effects on survival and resilience have not been demonstrated [54]. Therefore, it is possible that, in the present study, mothers who were unprepared lacked knowledge about the items that should be included in a grab bag. Additionally, they may have been unconvinced of the effectiveness of a grab bag and, thus, unwilling to spend their time preparing one. In light of this, it may be necessary to identify the supplies that children and their mothers require for survival and recovery and to ensure that mothers are prepared to take such supplies with them in an emergency rather than simply having the recommended grab bag. The parents of children with food allergies have been shown to regularly experience high levels of emotional distress and anxiety [55]. Considering the high frequency of heavy rainfall disasters in Japan in recent years due to global warming [56] and the increasing prevalence of food allergies in preschool children in developed countries [57,58], it is important to prepare for the evacuation of children with allergies and to create an evacuation environment where mothers and their children can feel safe. In addition, it is necessary to create an environment where the mothers of children with allergies can receive support from others during disasters. Medically vulnerable populations are more susceptible to the effects of disasters; it is therefore important to increase mutual assistance for these populations in normal times in order to reduce stress during disasters and evacuations [59]. In Japan, the lessons learned from the Great Hanshin-Awaji Earthquake and the Great East Japan Earthquake have shown that community-level mutual assistance during normal times helps people to resume their lives after disasters and reduces the physical and mental effects of evacuation [60,61]. The emergency response to the COVID-19 pandemic, which was a global disaster, also emphasized the importance of community-level mutual assistance [62]. Both these previous findings and the present results indicate the necessity of identifying the mothers of children with allergies who lack support from their neighbors and taking action to redress this. Mothers and their children have mutual effects on one another [14,19], so if mothers are anxious about the effects of a disaster, this may have a negative effect on their children. Therefore, it is necessary to prepare disaster supplies and build community-level mutual assistance for mothers, especially those with children under 3 years old, in accordance with their children’s physical and mental state and allergies.

There were some limitations to this study. The target population was a childcare center in one municipality of Ishikawa Prefecture. The timing of the survey encompassed part of both the summer and autumn seasons, and so the risk of disasters varied over this period. In addition, we were unable to obtain data on the prevalence and treatment of chronic diseases other than allergies. This may have affected the responses. Regarding disaster preparedness for children with allergies, the mothers of children with newly diagnosed food allergies may have planned to adopt measures to address the allergies but had not done so before the survey. This was a cross-sectional study, so it was not possible to establish causal relationships between the variables under investigation. Despite these limitations, our study provides some considerations for policymakers in developing evacuation environments in response to disasters. The collection of data on chronic illnesses in children and concerns and anxieties in mothers is important in the preparation of shelter environments and the development of stockpiling strategies. It is also important to ensure that children with chronic illnesses are identified and that appropriate support measures are put in place during evacuation. Additional studies are needed in the future that investigate a larger number of target municipalities and that obtain more representative samples.

## 5. Conclusions

Municipal policymakers need to be aware that mothers with children younger than 3 years and those whose children who have allergies experience anxiety about the effects of evacuation on their children. Measures that identify children with allergies and provide allergen-specific provisions for evacuation shelters are recommended. For example, the stockpiling and delivery of allergy-free food are possible solutions to support evacuees with food allergies. In addition, the establishment of an environment where the mothers of children attending childcare centers can interact with other community members, share information about their children’s condition, and discuss countermeasures is recommended. In addition to fostering community-level connections and addressing mothers’ worries, the preparation of a disaster risk reduction grab bag tailored to each child’s pre-existing conditions and characteristics would be useful. A related recommendation is to prepare a minimum amount of supplies that are carefully selected, rather than a generic grab bag, to help children survive and recover in the event of an evacuation. However, the selection of such supplies must take into account the possibility that carrying more supplies may prevent a child from evacuating quickly.

## Figures and Tables

**Table 1 ijerph-20-01850-t001:** Demographic characteristics of participants (*n* = 1298).

Variables		*n*	%
Mothers’ age, mean (standard deviation)	34.7 (7.0)		
Number of years in current residence, mean (standard deviation)	5.9 (6.6)		
Pregnant	No	1219	93.9
	Yes	79	6.1
Mothers’ worries about the effects of evacuation on their children	No	583	44.9
	Yes	798	61.5
Number of children, mean (standard deviation) (*n* = 2543) *	2.0 (0.8)		
Children’s age (years), mean (standard deviation) (*n* = 2543) *	4.6 (3.5)		
Number of children per age group (*n* = 2543) *	0 years	174	6.8
	1 year	256	10.1
	2 years	337	13.2
	3 years	332	13.1
	4 years	344	13.5
	5 years	338	13.3
	≥6 years	762	30.0
Breastfeeding	No	1232	94.9
	Yes	66	5.1
Eats baby food	No	1228	94.6
	Yes	70	5.4
Some kind of allergy	No	1142	88.0
	Yes	156	12.0
Illnesses other than allergies	No	1249	96.2
	Yes	49	3.8

* Indicates the number of children whose mothers responded. Therefore, these values include children who did not attend childcare centers.

**Table 2 ijerph-20-01850-t002:** Types of allergies, allergens, and disaster preparedness (*n* = 1298).

Item	Category	*n*	%
Food allergies	No	1210	93.2
	Yes	88	6.8
Allergens (*n* = 88) (multiple answers)	Chicken egg	61	69.3
	Milk	17	19.3
	Wheat	7	8.0
	Buckwheat	5	5.7
	Peanuts	4	4.5
	Shrimp	1	1.1
	Crab	0	0.0
	Other	19	21.6
Use of allergy-free ingredients (*n* = 88)	No	37	42.0
	Yes	51	58.0
Stockpiles allergy-free foods (*n* = 51)	1–3 days	10	19.6
	1 week	4	7.8
	>2 weeks	1	2.0
	No	36	70.6
Medication for allergies (*n* = 156)	No	79	50.6
	Yes	77	49.4
Stockpiles medicine (*n* = 77)	1–3 days	10	13.0
	1 week	22	28.6
	>2 weeks	39	50.6
	No	6	7.8
Atopic dermatitis	No	1206	92.9
	Yes	92	7.1
Treatment (multiple answers) (*n* = 92)	Moisturizing lotion	41	44.5
	Steroid ointment	49	53.2
	Other	5	5.4

**Table 3 ijerph-20-01850-t003:** Mothers’ and children’s backgrounds and disaster preparedness in relation to mothers’ worries about effects of evacuation on their children (*n* = 1298).

Item				Mothers’ Worries about the Effects of Evacuation on Their Children				*p*-Value
	Category	Total		No		Yes		
		*n*	%	*n*	%	*n*	%	
Mothers’ background								
Age	<35 years	607	46.8	232	38.2	375	61.8	0.86
	≥35 years	691	53.2	268	38.8	423	61.2	
Pregnant	No	1219	93.9	469	38.5	750	61.5	0.90
	Yes	79	6.1	31	39.2	48	60.8	
Number of years in current residence	<5 years	860	66.3	319	37.1	541	62.9	0.14
	≥5 years	438	33.7	181	41.3	257	58.7	
Experience of evacuation	No	1275	98.2	492	38.6	783	61.4	0.83
	Yes	23	1.8	8	34.8	15	65.2	
Disaster preparedness and supporters								
Can visualize living in an evacuation shelter	No	1198	92.3	457	38.1	741	61.9	0.39
	Yes	100	7.7	43	43.0	57	57.0	
Have prepared a grab bag for disaster risk reduction	No	1033	79.6	416	40.3	917	88.8	0.01
	Yes	265	20.4	84	31.7	181	68.3	
Have parents/relatives who can help in case of a disaster	No	330	25.4	126	38.2	204	61.8	0.89
	Yes	968	74.6	374	38.6	594	61.4	
Have neighbors who can help in case of a disaster	No	609	46.9	199	32.7	410	67.3	0.00
	Yes	689	53.1	301	43.7	388	56.3	
Children’s background								
Children’s sex	Male	659	50.8	250	37.9	409	62.1	0.69
	Female	639	49.2	250	39.1	389	60.9	
Number of children	1	385	29.7	120	31.2	265	68.8	0.00
	≥2	913	70.3	380	41.6	533	58.4	
Children’s age	<3 years	601	46.3	201	33.4	400	66.6	0.00
	≥3 years	697	53.7	299	42.9	398	57.1	
Breastfeeding	No	1232	94.9	485	39.4	747	60.6	0.00
	Yes	66	5.1	15	22.7	51	77.3	
Eats baby food	No	1228	94.6	486	39.6	742	60.4	0.00
	Yes	70	5.4	14	20.0	56	80.0	
Some kind of allergy	No	1142	88.0	466	40.8	676	59.2	0.00
	Yes	156	12.0	34	21.8	122	78.2	
Illnesses other than allergies	No	1249	96.2	489	39.2	760	60.8	0.02
	Yes	49	3.8	11	22.4	38	77.6	

**Table 4 ijerph-20-01850-t004:** Factors related to mothers’ worries about the effects of evacuation on their children (*n* = 1298).

Item	Category	Mothers’ Worries about the Effects of Evacuation on Their Children			
		OR	95% CI		*p*-Value
			Lower Limit	Upper Limit	
Mothers’ age (years)	<35/≥35 years	1.139	0.898	1.445	0.282
Have prepared a grab bag	No/Yes	1.494	1.112	2.008	0.008
Have neighbors who can help in case of a disaster	No/Yes	1.559	1.223	1.988	0.000
Number of children	1/≥2	1.286	0.976	1.694	0.074
Children’s age	<3/≥3 years	1.333	1.036	1.715	0.025
Breastfeeding	Yes/No	1.317	0.645	2.688	0.450
Eats baby food	Yes/No	1.788	0.867	3.685	0.116
Some kind of allergy	Yes/No	2.302	1.530	3.461	0.000
Diseases other than allergies	Yes/No	1.888	0.933	3.820	0.077

Binomial logistic regression analysis; abbreviations: CI: confidence interval, OR: odds ratio.

## Data Availability

The data analyzed during this study are included in this published article. Further inquiries can be directed to the corresponding authors.

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
