# Peer review of "Understanding Mothers’ Worries about the Effects of Disaster Evacuation on Their Children: A Cross-Sectional Study"

_ijerph, 2023, doi:10.3390/ijerph20031850_

Round 1

Reviewer 1 Report

Abstract

- No need to state all the numbers from the Results.

Introduction

- Line 41: can you please elaborate this statement?

- Did you have any hypotheses?

2.2. Data collections:

- Please consider joining all the subsections into one and write the info in paragraphs.

- Line 139: why these exact ages?

- Lines 143-143: it would be better to use continuous variables. Could you please explain why you performed the data collection differently?

- Lines 152-153: what do you mean with dependent and independent variables? Maybe you mean that you used predictors and criteria for the regression analysis?

Results

- Lines 177-186 & Table 1: please do not repeat the results (in text and in the table). Use one option only (I believe the table is more suitable one; then you just prepare a short description).

- Lines 194-199 & Table 2; Lines 208-210 & Table 3: same.

- Table 2: could you please edit the table a bit? It is very large with large gaps that could be reduced.

Reviewer 2 Report

This article tries to identify the main concerns of those Japanese mothers who are at-risk of experiencing a natural disaster. The analysis is very poor and limited, and I have serious doubts about its capacity of being really informative. Here my major points:

1) Paragraph 2.1 contains very few info barely correlated and apparently random. Why do you think that specific geographical statistics are important for these article? Shouldn't you include something more related maybe to families, children or women?

2) I admit to have almost no knowledge about the framework of this paper, but I really not understand why did you put so much emphasis on allergies. Why did not you asked something about - I don't know - disabilities or chronic diseases? Why allergies are so important for escaping from a natural disaster? And why the only disease not related with allergies is dermatitis?

3) You should compute a descriptive statistics table. It is very difficult to understand your sample composition. Some information is written in the text, some other is inferable from the results table. A descriptive statistics table is available only for allergies, for no good reason. Moreover, the reader is not so interested in absolute numbers, especially if numbers are included in the text and not in apposite tables. Please, just refer on percentages, unless you compute a new table.

4) At line 144 you refer to some childcare guidelines. Could you please clarify what are them about?

5) I really struggled to understand your dependent variables. If I got it right, you employed the dichotomic answer to the question "Are you worried about the effect that disaster evacuation will have on your children?". If my guess is correct, please make this clear in the article. Otherwise, you should clarify how that variable is computed. Moreover, you need you provide how many mothers are worried and how many are not. If I infered correctly from Table 2, more or less 38.6% of your sample is not worried but it is just what I understood. This crucial information is never stated extensively. Finally, you should discuss what this variable means. Namely, what is the feeling captured from this question, in your opinion? In a country such as Japan, which is at high seismic risk, does it make sense to argue that someone could be not worried for their children? What should be the line between being worried or not?

6) What does it mean that mother's age is used as a confounding factor?

7) Are you sure that forcing the entry of multicollinear variables is the right choice? At the end you just come with an overfitted model. Which variables are at-risk of being multicollinear? Can they be collapsed in a single information, maybe?

Reviewer 3 Report

The detailed manuscript review report is attached herewith.

Author Response

Please see the attachment,

Reviewer 4 Report

The authors presented a well-written manuscript describing their single-center cross-sectional study in Ishikawa Prefecture, Japan. They explored the main factors causing anxiety to mothers in case of an evacuation.

The data is presented in a clear and straightforward format and the results were adequately analysed and explained.

The authors correctly mention that medically vulnerable populations are more susceptible to the effects of disasters. It would be interesting to know how many children had other medical conditions besides allergies, as this might have also influenced their mother's answers to the questionnaire. If there is no data regarding this information, then it might be worth mentioning this potential confounding factor in the limitations section.

Please correct the typo on line 66: ...disaster supply kits...

Reviewer 5 Report

The psychological stress of mothers in a disaster evacuation is transmitted to their children, which can impact their children’s mental health. So, this study employed a questionnaire to identify factors (such as children’s allergies and disaster preparedness) associated with mothers’ maternal anxiety about the effects of disaster evacuation on children. The results suggest that mothers of children under 3 years of age with allergies who are unprepared and do not have social support are likely to be worried about the effect of evacuation on their children. The manuscript is interesting, in which the amount of sample data used for the statistic is reasonable, and the target area is representative, but some details still need to be revised.

1. I hope the authors will provide a more detailed explanation of the statistical analysis process and interpretation of statistical result indicators in lines 149-161.

2. The author mentioned in line 152, “Mothers’ worries about the effects of evacuation on their children was used as the dependent variable, and the independent variables were have not prepared a grab bag for disaster risk reduction”. There may be a problem with the formulation of the independent variable here, as there seem to be many independent variables like “have neighbors who can help in case of a disaster”. The author’s formulation may not be rigorous and is somewhat confusing.

3. There is no first-line indent for paragraphs in section 2.2. Please make a change.

4. I noticed that the two percentages in lines 173-174 add up to more than 100% and suggested that the authors explain this. “Of participants, 798 (61.5%) reported anxiety about the effects of disaster evacuation on their children, and 583 (44.9%) did not.”

5. In lines 267-274, the authors discuss the limitations of their study, such as the timing of the survey. I suggest that the authors present some outlooks in this section to go along with the limitations to show some directions for future research and make the discussion more complete.

6. I hope that the author will polish the language in the manuscript to make the sentences in the manuscript more concise in form. Some sentences in the manuscript have multiple uses of the preposition “of ”, which may make it difficult to read.

Round 2

Reviewer 2 Report

The answers to the points I addressed are now clear and exhaustive, so as the way they were included in the manuscript. I think the article can now be published.